# Evaluating dengue burden in Africa in passive fever surveillance and seroprevalence studies: protocol of field studies of the Dengue Vaccine Initiative

Jacqueline Kyungah Lim,[1,2] Mabel Carabali,[1,3] Jung-Seok Lee,[4] Kang-Sung Lee,[4] Suk Namkung,[1] Sl-Ki Lim,[1] Valéry Ridde,[5] Jose Fernandes,[6] Bertrand Lell,[6] Sultani Hadley Matendechero,[7] Meral Esen,[8] Esther Andia,[9] Noah Oyembo,[9] Ahmed Barro,[10] Emmanuel Bonnet,[11] Sammy M Njenga,[9] Selidji Todagbe Agnandji,[6] Seydou Yaro,[12] Neal Alexander,[2] In-Kyu Yoon[1]

For numbered affiliations see end of article.

**Correspondence to**
Jacqueline Kyungah Lim;
kalim@ivi.int

## ABSTRACT

**Introduction** Dengue is an important and well-documented public health problem in the Asia-Pacific and Latin American regions. However, in Africa, information on disease burden is limited to case reports and reports of sporadic outbreaks, thus hindering the implementation of public health actions for disease control. To gather evidence on the undocumented burden of dengue in Africa, epidemiological studies with standardised methods were launched in three locations in Africa.

**Methods and analysis** In 2014–2017, the Dengue Vaccine Initiative initiated field studies at three sites in Ouagadougou, Burkina Faso; Lambaréné, Gabon and Mombasa, Kenya to obtain comparable incidence data on dengue and assess its burden through standardised hospital-based surveillance and community-based serological methods. Multidisciplinary measurements of the burden of dengue were obtained through field studies that included passive facility-based fever surveillance, cost-of-illness surveys, serological surveys and healthcare utilisation surveys. All three sites conducted case detection using standardised procedures with uniform laboratory assays to diagnose dengue. Healthcare utilisation surveys were conducted to adjust population denominators in incidence calculations for differing healthcare seeking patterns. The fever surveillance data will allow calculation of age-specific incidence rates and comparison of symptomatic presentation between patients with dengue and non-dengue using multivariable logistic regression. Serological surveys assessed changes in immune status of cohorts of approximately 3000 randomly selected residents at each site at 6-month intervals. The age-stratified serosurvey data will allow calculation of seroprevalence and force of infection of dengue. Cost-of-illness evaluations were conducted among patients with acute dengue by Rapid Diagnostic Test.

**Ethics and dissemination** By standardising methods to evaluate dengue burden across several sites in Africa, these studies will generate evidence for dengue burden in Africa and data will be disseminated as publication in peer-review journals in 2018.

### Strengths and limitations of this study

► There have not been population-based studies conducted with a multidisciplinary approach (ie, surveillance, healthcare utilisation and serosurvey in one catchment area population). Data from the passive surveillance will be used to calculate annual incidences of dengue and data from the serosurvey will estimate the force of infection and prevalence.

► The studies were conducted in three locations in Africa, based on standardised methods and laboratory algorithm. Thus, comparison by site would be possible.

► This is not a cohort study. The passive facility-based surveillance may lead to underestimation of the burden of dengue fever by measuring incidence based on only those that sought care at our study facilities.

► There may be limited generalisability of our study results to other dengue-endemic parts of Africa.

## BACKGROUND

Dengue fever, a mosquito-borne flavivirus infection caused by four related but antigenically distinct dengue viruses (DENVs, serotypes 1–4), is a major and rapidly increasing global public health problem. Recent studies have estimated an annual incidence of 50–100 million symptomatic infections globally.[1] Dengue is a high burden disease that disproportionately affects countries in the tropics and subtropics, many of which have limited healthcare resources.[2] Although one dengue vaccine has been recently licensed in several endemic countries, the vaccine has restricted age and epidemiological indications. Other prevention and control measures such as vector control are suboptimal as

stand-alone interventions,[3 4] and no drugs for treatment are currently available.

Like in Asia and the Americas, epidemics of dengue were reported from Africa in the late 19th and early 20th centuries.[5 6] Specifically for Africa, there are records of multiple dengue case reports between 1964 and 1968 with DENV 2 in Nigeria.[7] Data from several studies conducted in the 1960–1970s in Nigeria supported a substantially high level of immunity in adults as well as children.[8 9] In 2011, Amarasinghe *et al* conducted a comprehensive review of literature on dengue in Africa and described that dengue cases have been reported in 34 countries in Africa, with most of these countries also having *Aedes* mosquitoes.[6] However, prior studies which suggested the presence of dengue in Africa were limited by their retrospective design or sample collection (blood donors or sample collected from surveys of other diseases), and often from travellers, with a small number of reported autochthonous cases, to demonstrate the true, population-based, burden of dengue. Also, while many dengue endemic countries in Asia and Latin America have mandatory reporting of dengue cases to public health authorities and national surveillance systems in place to monitor incidence patterns,[10] most African countries lack such established reporting mechanisms and only sporadic outbreaks and individual case reports have been documented. In addition, the frequently non-specific clinical presentation of dengue may be difficult to distinguish from the myriad other infectious diseases present in Africa, since dengue diagnostic assays are not widely available. Thus, the burden of dengue remains largely unknown in Africa.[6 11] Without such dengue burden data, informed decision-making about prevention and control measures, including dengue vaccine introduction, in Africa are not possible.

Limited by surveillance capacity hindering continuous reporting in the region, there had not been frequent and systematic reporting of dengue in Africa. African ancestry is known to be protective against severe dengue and the candidate genes were recently identified in a Cuban patient.[12 13] Bhatt *et al*'s modelling of the global dengue burden suggests high burden in Africa in terms of equal numbers of infections (both apparent and inapparent) as in Latin America.[1] There are new findings about dengue in Africa, but there is still much unknown about the magnitude of the dengue problem in the continent. To improve estimates of population-based dengue disease burden in Africa and validate whether the undocumented burden of dengue is as high in Africa as in the Americas with empirical data, the Dengue Vaccine Initiative (DVI) initiated field studies at three sites in West (Ouagadougou, Burkina Faso), West-Central (Lambaréné, Gabon) and East Africa (Mombasa, Kenya). In each of the three sites, a standardised package of study components, including passive facility-based fever surveillance, healthcare utilisation surveys, cost-of-illness surveys and serological surveys (figure 1), was initiated between December 2014 and March 2016.

## METHODS
### Site selection
Study sites were selected, in part, based on their likelihood of supporting DENV transmission. To select sites, we considered dengue outbreaks and cases reports in the literature, available seroprevalence studies as well as country-specific dengue risk maps of the probability of DENV transmission and the level of evidence of dengue presence, reporting the uncertainty of the consensus estimates of dengue in Africa.[7 14] In addition, adequate research infrastructure to implement the studies was taken into account. Finally, inclusion of different regions of Africa was also a factor in site selection. Thus, Ouagadougou, Burkina Faso; Lambaréné, Gabon and Mombasa, Kenya were selected, respectively, to measure the burden of dengue in selected sites from West, (West-) Central and East Africa.

In Burkina Faso, the first reported dengue outbreak occurred in Ouagadougou in 1982 due to DENV-2.[6] Serological prevalence of dengue antibodies among pregnant women and blood donors was found to be 26.3% in a rural setting (Nouna village) and 36.5% in an urban setting (Ouagadougou) in 2006.[15] More recently, an observational study conducted by Ridde *et al* among febrile patients consulting at selected study facilities in 2013–2014 showed 8.7% (33/379) to be positive by dengue rapid diagnostic test (RDT) and 15 of 60 samples tested by RT-PCR to be dengue-positive.[16] With evidence for the presence of dengue, along with a strong health and demographic surveillance system (Ouaga-HDSS) which could be used to describe the demographic characteristics of the catchment area, a field study was initiated in Ouagadougou, Burkina Faso in December 2014.

In Gabon, cases of dengue haemorrhagic fever (DHF) caused by up to three different DENV serotypes have been reported, and dengue seroprevalence has been found to be between 5% and 20%.[17–19] Results of a recently published study demonstrated seroprevalence of 12.3% among toddlers approximately 30 months of age in semirural Lambaréné between 2007 and 2010.[20] However, a different study in 2005–2008 suggested minimal DENV transmission in rural areas of Gabon.[21] This latter study examined antibodies against dengue in individuals from randomly selected villages representing about 10% of all Gabonese villages. Blood samples were tested by anti-DENV IgG and IgM capture ELISA and found to have only minimal IgG (0.5%) and IgM (0.5%) seroprevalence. Based on these low prevalences, the authors concluded that there was no active circulation of DENV in rural Gabon. However, the low seroprevalence may have been affected by low sensitivities of the tests used, leading to a high rate of false negative results and/or selection bias in the blood sample pool among the selected villagers.[22] Seroprevalence estimates in the 2007/2010 study may have also been impacted by the possibility of false-positive results due to IgG cross-reactivity among flaviviruses.[21] Nevertheless, given the

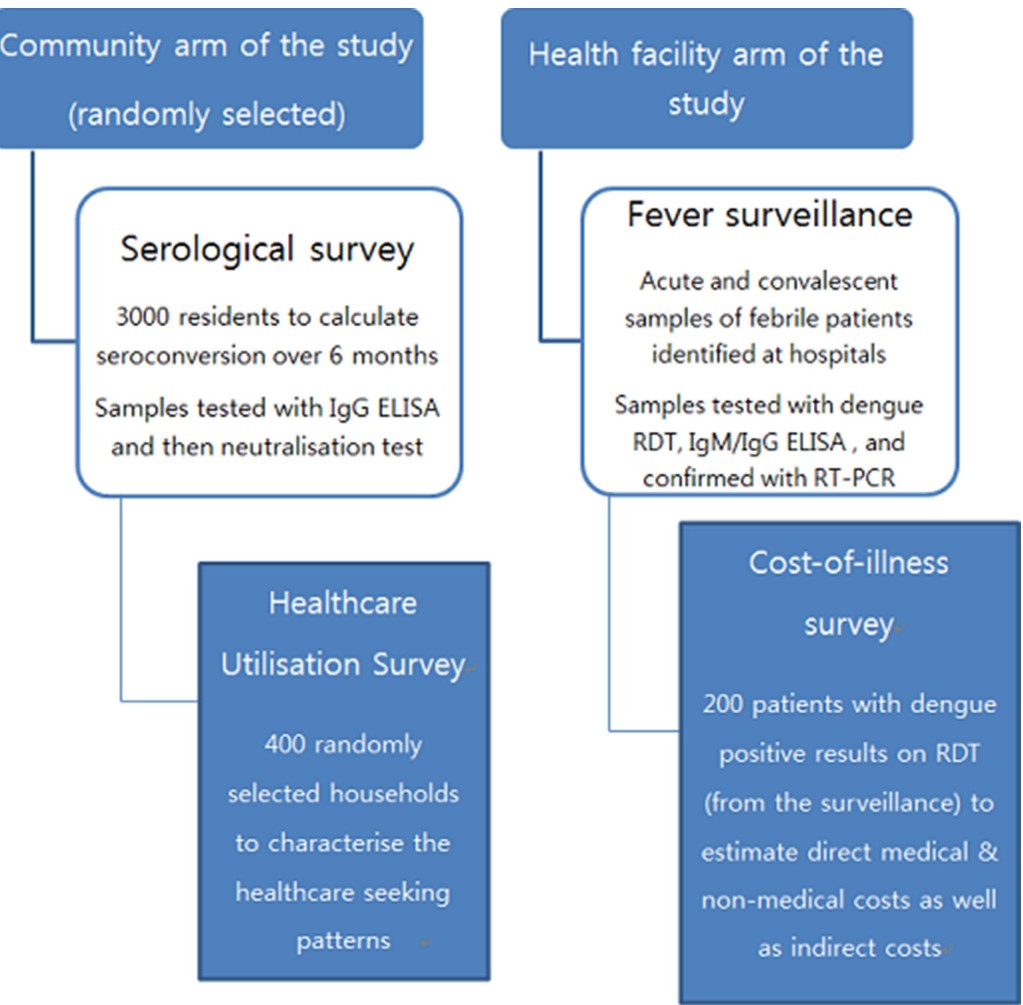

**Figure 1** Description of the study components, including passive facility-based fever surveillance, healthcare utilisation surveys, cost-of-illness surveys and serological surveys. There are two arms in the study package, composed of four parts. In the health facility-based arm of the study package, there are passive facility-based fever surveillance and cost-of-illness survey embedded within the surveillance. In the community arm of the study, there are serological survey and healthcare utilisation survey.

possibility of DENV circulation in Gabon, a field study was initiated in Lambaréné in March 2015 in a community with a catchment population of about 77 000 residents, using the clinical research infrastructure of the Centre de Recherches Medicales de Lambaréné (CERMEL), benefiting from experienced research staff who conducted a large Phase 3 malaria vaccine trial.[23 24]

In Kenya, more evidence is available for the presence of dengue based on local data. Dengue was the most common viral pathogen in retrospectively tested blood specimens from HIV-negative survey samples from the 2007 Kenya AIDS Indicator Survey. Antibody testing for dengue as well as chikungunya and Rift Valley fever was performed by IgG ELISA using either commercial kits or CDC assays; 12.5% were found to be dengue-positive.[25] Similarly, a household survey found 13% of individuals from 701 households in Mombasa had serological evidence of either past or current DENV infection.[26] These data suggest that there is more dengue in Kenya than indicated by public health reporting, possibly due to misdiagnosis.[25 26] A field study was initiated in Mombasa, Kenya in March 2016.

### Study participants
For the passive facility-based fever surveillance, individuals who met the following criteria were eligible for study enrolment:
1. Age 1–55 years old.
2. Resident of the catchment area covered by healthcare facilities participating in the study, without plans to move out of the catchment area within 12 months.
3. Signed informed consent and assent for those aged between 7 (13 for Kenya) and 17 years.
4. Patients presenting with current fever (axillary temperature ≥37.5°C) or history of fever for ≤7 days duration without localising signs (fever caused by a localised infection as well as fever with a known and confirmed aetiology other than dengue, such as malaria

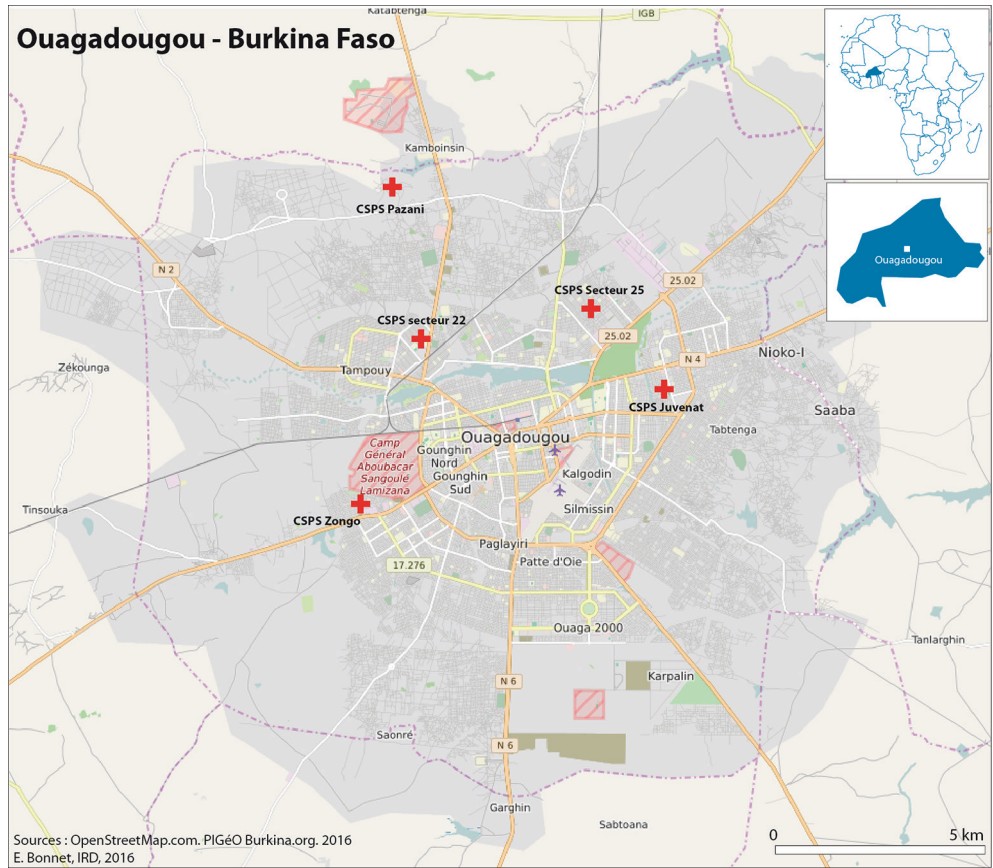

**Figure 2** Map of the study area in Ouagadougou, Burkina Faso.

confirmed by malaria RDT, as listed in the patient identification standard operating procedure [SOP]).

For the serological survey, criteria 1–3 were applied. For the healthcare utilisation survey, household interviews were conducted among the heads or representatives of the household invited from each family participating in the serosurvey.

### Study area and population

Burkina Faso, located in West Africa, has a population of 14 017 462. The country is mainly rural with about 29% of the population reported to be living in urban areas in 2014. However, Burkina Faso is urbanising rapidly and is positioned as the country with the fourth fastest urbanisation in the last 25 years.[27 28] The capital, Ouagadougou, has a population of 2 741 128. The majority of the population live in urban settings. About 45% of the population are under 15 years of age.[29] The city is divided into 12 districts and 52 sectors. Ouagadougou is the country's largest city and the cultural and economic centre. The city is part of the Soudano-Sahelian area, with a rainfall of about 800 mm per year. The rainy season is from May to October, with a mean temperature of 28°C (82°F). The cold season runs from December to January, with a minimum average temperature of 16°C (61°F). During the hot season, which runs from March to May, the temperature can reach as high as 43°C (109°F).

The HDSS is in place in Ouagadougou. Ouaga-HDSS monitors a population of 81 717 residents; according to this surveillance system, the city population is very stable with a rate of migration of 4.1% and more than 80% of the inhabitants with ownership of their houses [20]. A map of the city and the study area is shown in figure 2.

Gabon, located on the west coast of Central Africa, has an area of nearly 270 000 square kilometres (100 000 sq. mi) with a population estimated at 1.5 million. Its capital and largest city is Libreville. In 2014, it is reported that 87% of the Gabonese population lived in urban areas.[28] The sixth largest city, Lambaréné, the capital of Moyen-Ogooué province, is located 75 km south of the equator, with a population of 25 257 in 2009. The majority of Lambaréné residents live in semirural areas. About 42% of the Gabonese population is under 15 years of age.[29] Similarly, Lambaréné's population is relatively young with about 50% under 20 years of age.

The health services of Gabon are mostly public, but there are some private institutions as well. With one of the best medical infrastructure in the region, almost 90% of the population have access to healthcare services. Albert Schweitzer Hospital (ASH) is a private institution which served as a study site for the passive fever surveillance study.[30 31] The study area in Lambaréné is shown in figure 3.

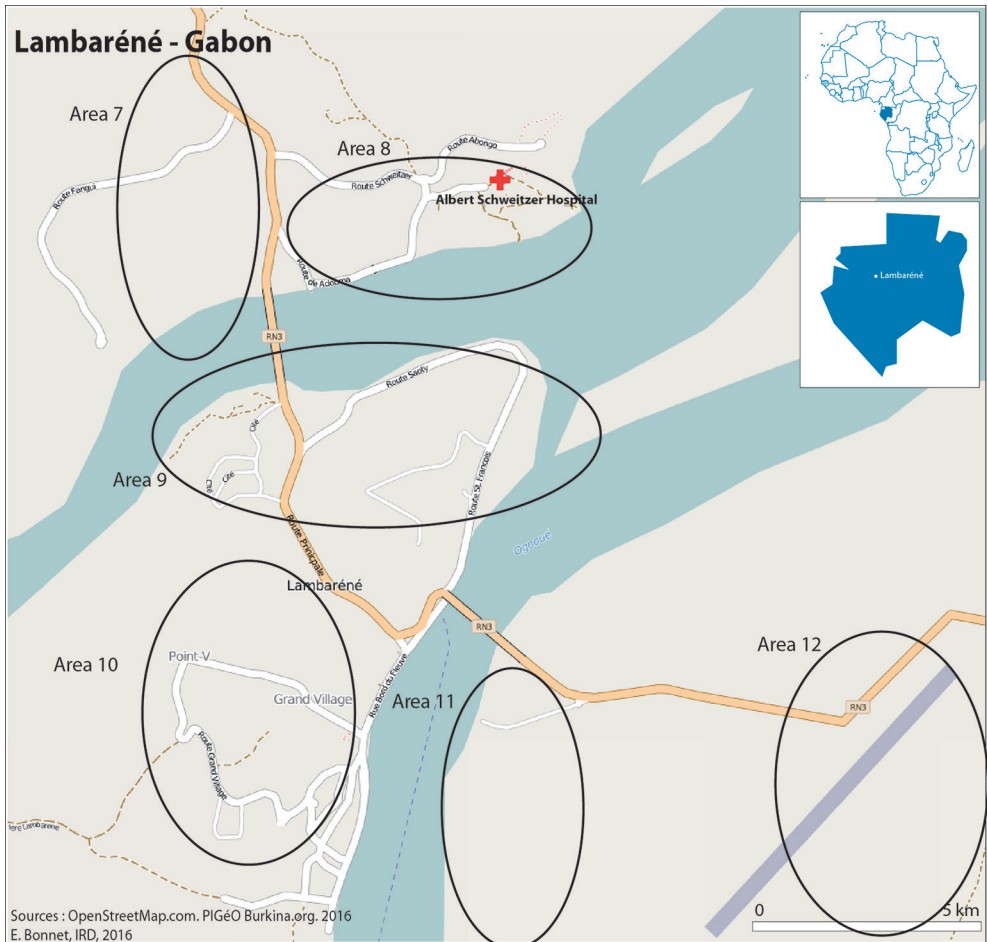

**Figure 3** Map of the study area in Lambaréné, Gabon.

Kenya, located in East Africa, lies on the equator, covering $581\,309\,\mathrm{km}^2$ ($224\,445$ sq. mi), with a population of approximately $45$ million people in 2014.[32] Kenya generally has a warm and humid tropical climate but is diverse, ranging from the cooler climate around the capital city, Nairobi, to a hot and dry climate inland as well as a desert-like climate in the north-eastern regions along the border with Somalia and Ethiopia.[32] The capital, Nairobi, is a regional commercial hub. The main industries include agriculture, exporting tea and coffee as well as the service industry.

Kenya is divided into 47 semiautonomous counties. Mombasa is the country's second largest city after Nairobi and is located on the east coast of the country.[32] Administratively, Mombasa is the capital of Mombasa County, which was formerly called Coast Province. This overall Coast region covers over $80\,000\,\mathrm{km}^2$ in the south-eastern part of Kenya, constituting about 15% of the country's land area, with a population of $3\,325\,307$ residents.

The main economic driver of Mombasa is tourism and trading industry. Mombasa itself has a population of about 1.3 million with almost 50% of the population under 15 years of age.[29] Increasingly, the population of the province lives in urban areas; at present about 45% live in Mombasa and other urban centres. The 'long rains' period begins around April and the 'short rains'

period begins in October.[32] Mean annual temperature ranges from 24°C to 27°C, but maximum temperature averages over 30°C between January and April.

Figure 4 shows the area of Mvita subcounty of Mombasa, which was the catchment area for the study in Kenya, with a catchment population of $74\,735$ residents. The map indicates the three facilities involved in the study.

### Sample size

Given the paucity of available age-specific dengue incidence data in the study countries or nearby countries, it was difficult to obtain population-based incidence to make assumptions when calculating sample sizes. The required catchment population for the passive facility-based fever surveillance was roughly estimated based on the limited data available in the literature. Annual incidence estimates were calculated based on available prevalence estimates with the assumption that the outcome of interest has zero prevalence at age zero, and that force of infection is constant. It was assumed that prevalence estimates found for one particular age group would be adjusted as the annual incidence and used across all ages.

Wichmann *et al* calculated an expansion factor for children by comparing data from three cohort studies to national surveillance data in Southeast Asia.[33] For children in Thailand, the age-specific expansion factors

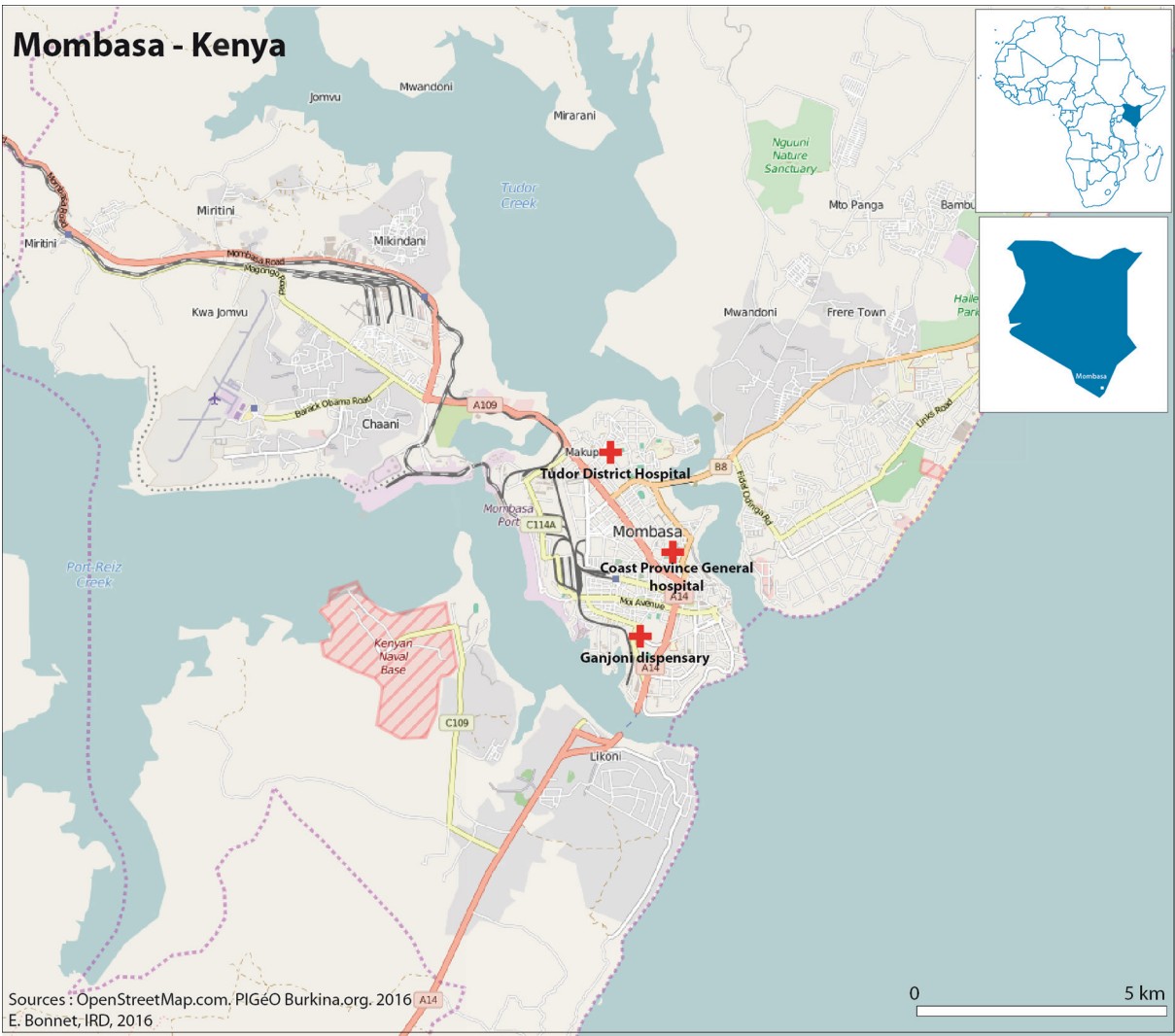

**Figure 4** Map of the study area in Mombasa, Kenya.

calculated were 11.85 for <5 years, 8.76 for 5–9 years and 7.81 for 10–14 years.[33] The results show that, even for Asia where better reporting and surveillance systems are available, there is a considerable degree of under-reporting. For Africa, there may be more dengue cases under-ascertained (not seeking care) and under-reported (not reported even if a patient with dengue seeks care, given that dengue is not one of the routinely notifiable diseases in Africa), but such information on the extent of underestimation of dengue was not available.[34 35] Also, the incidence estimates used in our sample size calculations were not from population-based studies. While it would have been ideal to adjust the incidence further for likely underestimation, the annual incidence used in sample size calculations could not be adjusted for possible under-reporting due to the lack of data. The sample sizes were calculated with 95% confidence levels and a margin of error at a fixed significance level within 25% of the true proportion of incidence. This gives relative precision of 75%, considering the gap in evidence for dengue incidence in the study areas. The final sample sizes calculated by assuming 10%–20% (variable by site) non-response rate or loss to follow-up. The required catchment population size for the fever surveillance study in Burkina Faso was estimated to be 100 000, Gabon to be 77 000 and Kenya to be 70 000. In these catchment populations, the number of enrolled subjects depends on the number of eligible patients who seek care at the study facilities. How many eligible febrile episodes would actually present at our study facilities was difficult to predict; but after assessment of the volume of febrile patients at the facilities, a realistic upper limit for enrolment for a study period of approximately 1.5 years was set at 3000 subjects to offer enrolment to all consenting eligible patients.

For the serological survey, the sample size was calculated similarly using the prevalence proportion based on published literature. Seroprevalence of 0.304 for Burkina Faso,[15] 0.123 for Gabon,[21] and 0.144 for Kenya[36] were used. With the same confidence levels and allowed margin of error and assuming 10%–30% (variable by site) non-response rate, the sample size was calculated to be 3000 participants at each site. Again, with the scarcity of data from the selected countries, there were no other

prevalence estimates reported or estimates from different age groups. As prevalence is expected to increase with age and higher prevalence would give a smaller sample size, our calculations are likely to be conservative.

## Study components

### Fever surveillance—design and methods

To determine burden due to symptomatic dengue in each of the three sites in Burkina Faso, Gabon and Kenya, passive facility-based fever surveillance was implemented in a well-defined catchment area population. In Burkina Faso, the surveillance study was initiated in December 2014 in five selected primary healthcare centres, locally called 'Centre de Santé et de Promotion Sociale', in the municipality of Ouagadougou, with a catchment population of 105 000 residents. This project was implemented in collaboration with Centre Muraz in Bobo-Dioulasso, EQUITE sante programme (a collaborative programme between University of Montreal and Action-Gouvernance-Integration-Reinforcement, AGIR, based in Ouagadougou, funded by the Canadian Institute of Health Research) and DVI. In Gabon, the surveillance study was initiated in the ASH serving a catchment population of 130 000 residents in the Moyen-Ogooué and surroundings within Lambaréné, in collaboration with CERMEL and Institute of Tropical Medicine in Tubingen, Germany. In Kenya, the surveillance study was implemented at Ganjoni dispensary, Tudor subcounty Hospital and Coast Provincial General Hospital, serving a catchment population of 70 000 residents in Mombasa, in collaboration with Kenya Medical Research Institute and Ministry of Health of Kenya.

As described in figure 5, both outpatients and inpatients at the designated study facilities, who meet inclusion criteria as mentioned earlier were tested for dengue, first with SD Dengue Duo RDT. Dengue confirmation was done by detection of dengue virus in serum samples using PCR as well as antidengue IgM and IgG antibodies in acute and convalescent serum by ELISA (SD Dengue IgM & IgG capture ELISA tests, Standard Diagnostics, Yongin-Si, Korea).[10 37] Every consecutive patient meeting inclusion criteria was eligible for enrolment during the study period. Infants<1 year old were not included due to operational limitations, such as difficulty of infantile bleeding.

In Ouagadougou, Burkina Faso, the fever surveillance was initiated in December 2014 and continued until February 2017 (approximately 2 years). In Lambaréné, Gabon, the fever surveillance was initiated in April 2015 and continued until January 2017 (approximately 1.5 years). In Mombasa, Kenya, the fever surveillance was initiated in March 2016 and continued until May 2017 (15 months).

Among subjects enrolled in the fever surveillance, those who were positive by dengue rapid diagnostic test were offered further enrolment in the cost-of-illness survey, consisting of interviews on the day of acute illness visit, day 10–14 from the first visit and day 28, if illness continues. The cost-of-illness survey questionnaire was designed to estimate the direct medical, direct non-medical and indirect costs associated with dengue-positive patients identified at study facilities. This survey also estimates the cost of treating dengue at the facility level. Data were gathered by linking patients' medical records concerning outpatient visits, inpatient visits and service consumption (eg, diagnostic tests, medication and other services provided to patients). The cost-of-illness portion of the study will be described separately.

### Fever surveillance—laboratory testing

As shown in figure 6, in all three sites, acute samples were tested using a commercial RDT for dengue NS1 and IgM/IgG (Dengue Duo, Standard Diagnostics, Yongin-Si, Korea). Dengue Duo RDT was used on the day of acute illness visit at the site of patient presentation (day 1). The acute and convalescent samples were subsequently tested at a local laboratory using dengue IgM/IgG ELISA (SD Dengue IgM & IgG Capture ELISA, Standard Diagnostics, Yongin-Si, Korea). The serum was separated and stored in 4 aliquots of about 500 µL for various laboratory tests, as indicated in consent documents.

After ELISA testing, samples were shipped to the International Vaccine Institute (IVI) in Korea. Samples with positive results by RDT or ELISA, as well as a small number of samples with negative results, undergo further testing by RT-PCR at the Clinical Immunology Laboratory of IVI. Four DENV serotype-specific real-time RT-PCR assays are used for laboratory confirmation of dengue and serotyping.[38] The DENV 1–4 RT-PCR assays are carried out in 25 µL reaction mixtures containing 5 µL template RNA, TaqMan Fast Virus 1-step mastermix (Applied Biosystems), 0.9 µM of each primer and 0.2 µM probe.[38] Amplification and detection are performed in a StepOne Plus real-time PCR system, and the baseline and threshold are determined using the auto-baseline and threshold feature in StepOne Software V.2.2.2 (Applied Biosystems). Thermocycling parameters are as follows: reverse transcription at 50°C for 5 min, inactivation at 95°C for 20 s, followed by 45 cycles of fluorescence detection at 95°C for 3 s and annealing at 60°C for 30 s.[38] A specimen is considered positive if target amplification is recorded within 40 cycles.

### Serological survey—design and methods

While the facility-based fever surveillance studies provide estimates of the burden of medically attended dengue disease, evaluation of all DENV infections in a population—including subclinical and mildly symptomatic infections, which impact immune status—is needed to capture the overall impact of dengue. As part of the study package, population-based serological surveys were conducted in the same catchment population used for the fever surveillance. At each of the three sites in Africa, the serosurvey was conducted on a cohort of approximately 3000 randomly selected residents of urban and semiurban parts of Ouagadougou, Lambaréné and

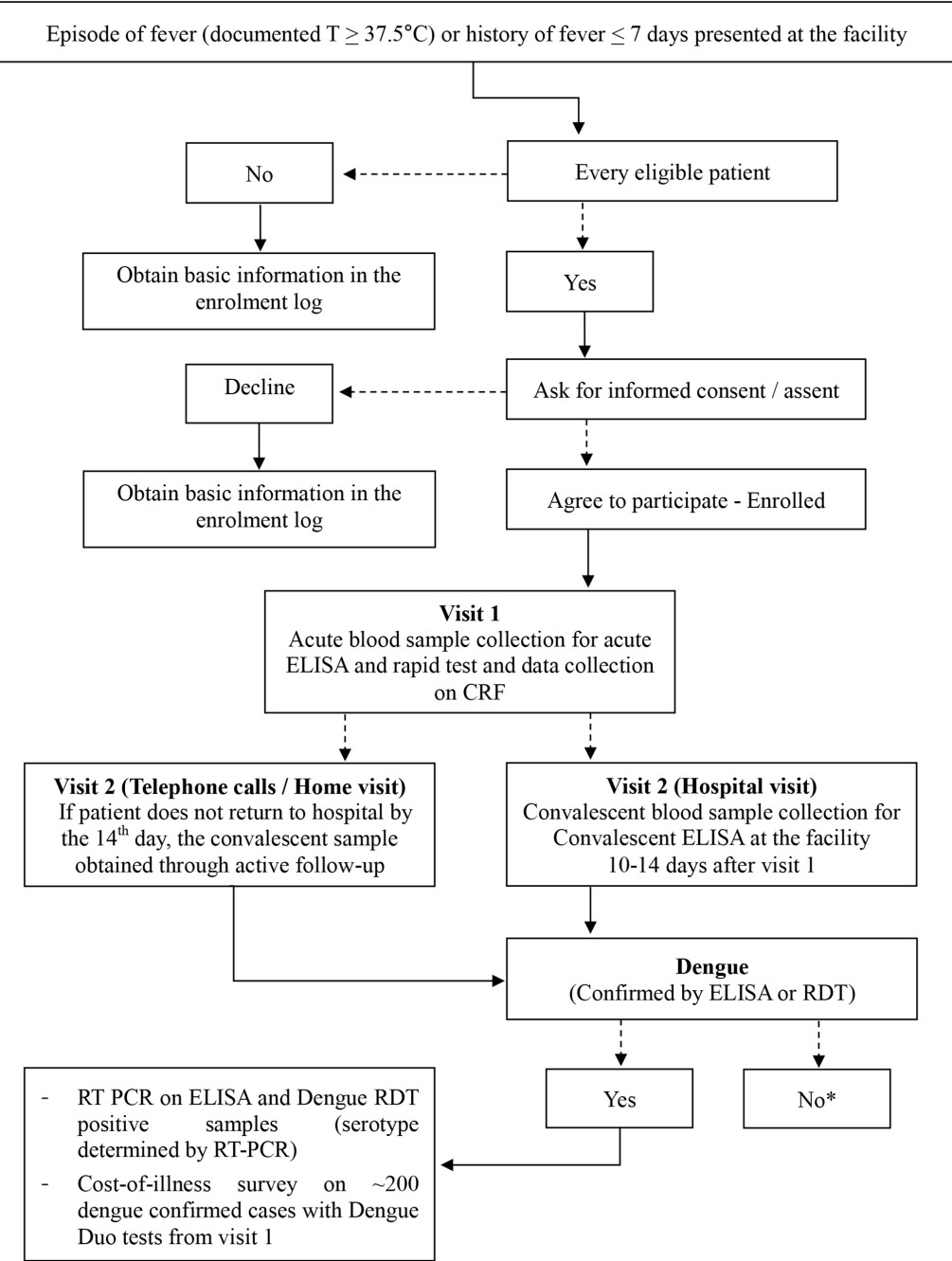

**Figure 5** Patient flow in the fever surveillance. Eligible febrile patients identified and enrolled as study subjects followed these steps to complete participation in the passive fever surveillance. * A small number of those samples that are negative on ELISA or NS1 are tested with PCR to exclude false negative results of the ELISA. CRF, case report form.

Mombasa. Without individual-level census information on all residents of Lambaréné and Mombasa, with help of community/village health workers, randomisation was done based on neighbourhoods (or defined areas for which the health workers/volunteers are responsible) as cluster units. As the community/village health workers are familiar with the villages and their residents, they are good entry points into the communities. With these health workers, the field team screened houses in the selected villages by knocking on doors of every 5–7 houses, depending on the household density per neighbourhood. Also, demographic information collected in

previous research projects conducted in the same area was used as a guide, if available. In the case of the site in Ouagadougou, HDSS data were available and the EQUITE SANTE, a CIHR funded research programme of the University of Montreal, had set up a geographic information system database of houses in the study area. Using these data, households of potential enrolees of the serosurvey were preselected randomly and household visits were made in Ouagadougou. In the three sites, about 45% of the serosurvey samples were targeted to be collected from children 1 to 14 years of age, and 55% were targeted to be collected from adults between 15 and 55 years of

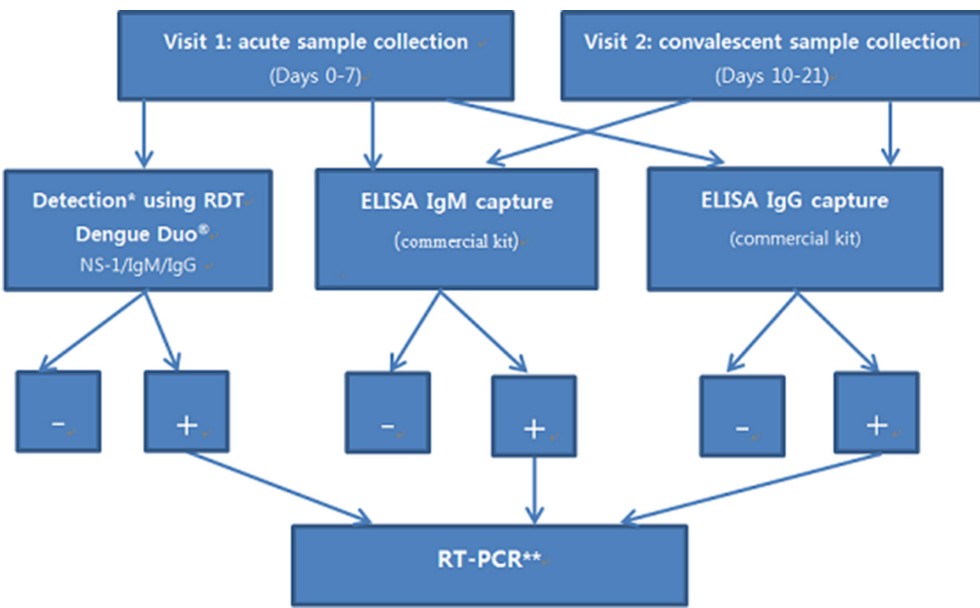

**Figure 6** Laboratory testing algorithm for dengue. Samples from subjects of the passive fever surveillance would follow these steps of the testing algorithm for confirmation of dengue. *Dengue Duo®test is performed on enrolled febrile patients to identify dengue cases for immediate follow-up of dengue-confirmed cases in the cost-of-illness survey. **Selected samples, including those that were found positive by IgM and NS1 on Dengue Duo®,as well as those positive by IgM and IgG capture ELISA, will be tested with RT-PCR.

age to reflect the age distribution of the general population of the area. Household-based enrolment was offered to the head of the household until the specific cap for the age-group was reached in Lambaréné and Mombasa.

Randomly selected subjects 1–55 years of age underwent phlebotomy (5 mL for children and 7 mL for adults) twice—before the rainy season and after the rainy season, at approximately 6-month intervals. The sera were evaluated using IgG indirect ELISA at baseline and after 6 months. The presence of dengue IgG antibodies at 6-month intervals will be used to estimate the level of occurrence of inapparent DENV infection and to calculate the rate of infection in the catchment population. Flow cytometry-based DENV neutralisation assays will be applied to a subset of samples to assess for presence of dengue neutralising antibodies and seroconversion over the 6-month interval. In addition to overall seroconversion, age-specific seroconversion estimates in the catchment population as well as the proportion of inapparent infections will be determined.

### Serological survey—laboratory testing

From the samples collected in the serosurvey, about 200 µL of serum were used and tested at a local laboratory using dengue IgG ELISA (Panbio Dengue IgG Indirect ELISA, Alere North America, Florida, USA). After ELISA testing for dengue IgG at the local laboratories, samples were shipped to IVI. Given potential serological cross-reactivity among flaviviruses,[39] flow cytometry-based neutralisation assays will be performed against selected flaviviruses to include yellow fever virus, West Nile virus, Zika virus and Japanese Encephalitis virus, in addition to DENV 1-4, at the Clinical Immunology Lab of IVI.[40 41]

About 50 samples per bleed for four bleeds in Burkina Faso and two bleeds in Gabon and Kenya will be tested.

About 1000 µL of serum is allotted for this procedure. The flow cytometry-based neutralisation assays are performed in duplicate in 96-well cell culture plates with flat-bottom wells, each containing DC-SIGN-expressing U937 cells.[40] The amount of virus used in the assay infects between 7% and 15% of the cells. Human immune sera are serially diluted and the virus is preincubated with the sera for 1 hour at 37°C.[40] The cells are washed, the virus and serum mixture is added to the cells for 1 hour at 37°C and the cells are further incubated for 24–48 hours at 37°C in 5% $CO_2$. The cells are fixed, permeabilised and stained with fluoresce-conjugated monoclonal antibody 4G2, which recognises the flavivirus E protein.[42] FACScan flow cytometer (Becton Dickinson, San Diego, California, USA) is used to analyse the cells.[40] The serum dilution that neutralises 50% of the viruses is calculated by nonlinear, dose-response regression analysis with Prism 4.0 software (GraphPad Software, San Diego, California, USA).

In addition, a Luminex-based multiplex immunoassay will be performed on a randomly selected subsample to assess for IgG to different flaviviruses.[43] About 200 samples per bleed for four bleeds in Burkina Faso and two bleeds in Gabon will be tested. Detection of IgG against ZIKV and each of the four DENV serotypes will be performed on patient serum samples using an in-house microsphere-based multiplex immunoassay (arbo-MIA) at the Clinical Immunology Lab of IVI.[44 45] The arbo-MIA is based on a mixture of microspheres covalently coupled with either DENV-1, DENV-2, DENV-3, DENV-4 or ZIKV

recombinant antigens (E protein domain III) produced in Drosophila S2 expression system. Briefly, microsphere mixtures were sequentially incubated in the dark under constant shaking with a 1:400 dilution of patient serum samples, with 2 µg/mL antihuman IgG biotin-conjugated antibody (Jackson Immunoresearch, West Grove, Pennsylvania, USA) and with 2 µg/mL streptavidin-R-phycoerythrin conjugate (Life technologies). After the final incubation, the median fluorescence intensity (MFI) of each microsphere set is quantified using a BioPlex 200 instrument (Bio-Rad Laboratories, Hercules, California, USA). Samples are considered seropositive if the ratio of MFI values obtained for the viral antigen to the control antigen is superior to the defined cut-off. The cut-off of the MIA is determined for each viral antigen by receiving operating characteristic (ROC) curve analysis using well-characterised sera.

In Lambaréné, the enrolment bleed took place in November–December 2015, while the second blood collection occurred in May 2016. In Ouagadougou, the enrolment bleed took place in May–June 2015 with follow-up blood collections in December 2015, June 2016 and January 2017. In Mombasa, the enrolment bleed took place in May 2016 with the second blood collection in November 2016–February 2017.

### Healthcare utilisation survey

As the passive fever surveillance was conducted at study facilities, patients with potential dengue could be missed if they seek care elsewhere. To identify the proportion of fever and dengue cases potentially missed by the passive surveillance system due to patients living in the study area but seeking care outside of study facilities, a population-based healthcare utilisation survey was conducted in 400 randomly selected households from the study catchment area to characterise the healthcare utilisation patterns of the households when they have (self-reported) febrile episodes among the family members. In addition to assessing health-seeking behaviours of the residents, preferences in terms of health-seeking behaviour and respective reasons for their preferences were investigated. The questionnaire was administered to 400 heads of households. Among 3000 residents who participated in the serosurvey, there were about 600 households. From these households, 400 heads of households were randomly selected and offered enrolment in the health utilisation survey. Heads of households or a senior representative within the household were asked questions on health seeking patterns of their family members.

### Study questionnaires

For the fever surveillance study, questionnaires were administered at the acute illness visit and the convalescent visit. The convalescent visit may take place at the healthcare facility (10–14 days later) or at the patient's home (15–21 days after the acute visit), according to patient preference and availability. The questionnaires were completed by medical staff of the study facilities, including demographic and clinical information (eg, signs, symptoms, past medical history, treatments prescribed and diagnoses). The same staff also completed the follow-up questionnaire at the convalescent visit within 21 days from the acute visit. Study nurses completed surveillance enrolment log. Lab technicians completed the lab section (mostly dengue-related diagnostics) and the forms were compiled by the study coordinator on site.

For the serosurvey component, questionnaires were administered at the household by trained field team staff at each serosurvey visit. Study nurses completed the questionnaire after a brief physical and medical examination. At the follow-up visit(s) in about 6 months, the same staff made the household visits to complete the follow-up questionnaire. Enrolment log was maintained by the study coordinator on site.

### Variables of the surveillance questionnaires

The variables collected are listed in table 1.

### Planned statistical analysis

From the fever surveillance data, incidence of symptomatic dengue among patients that seek healthcare at the study facilities will be calculated. Age-specific incidence rates in all the children and adults will be determined by referring to the size and distribution of the general population of the study area at the time of surveillance as the denominator in calculation of the incidence of symptomatic dengue cases. Each person residing in the study area is assumed to contribute 12 months of person time to the denominator. Although the study areas all report a low migration rate, the in-migration is assumed to balance the out-migration of the population during the study period. Age-specific incidence of symptomatic dengue will be calculated by using age-specific denominators and the number of symptomatic dengue cases in eligible individuals as the numerator.

Using the data collected in the Healthcare Utilisation Survey, the proportion of febrile cases missed by the passive surveillance system will be determined. Then using the proportion, the numerator will be further adjusted in recognition of those missed fever cases from the study area, which could have been dengue. Also, comparison will be made between those that agreed to participate and those that declined participation among the eligible potential enrolees. The enrolment log, which records basic information obtained during the screening process of potential enrolees, will be reviewed. In addition to checking that our sample of febrile cases is representative of febrile patients of the general population in the catchment area, refusal rates will be determined based on information in the log. Then, the refusal rates will be used to adjust the numerator.

SPSS software will be used for analysis of the fever surveillance data. Multivariable logistic regression will be used to compare confirmed patients with dengue versus patients with non-dengue febrile in terms of symptomatic presentation, based on signs and symptoms collected from all

**Table 1** List of variables collected in the passive fever surveillance data collection form

| Topic | Description | Items |
|---|---|---|
| Basic information | Demographic and basic information about the patient and the treatment received | Type of treatment, where patient is enrolled (IPD vs OPD)<br>Date of fever onset, duration of fever<br>Current temperature<br>Tourniquet test results<br>Patient's address (district and village-level)<br>Date of visit, date of birth, age and sex<br>Weight and height |
| General health condition | Current condition of the patient (self-report) and underlying diseases of the patient | How well the patient could handle daily activities<br>Pre-existing conditions |
| Signs and symptoms during this illness | A set of signs and symptoms that may be related to fever and dengue (dengue fever and dengue haemorrhagic fever) at both visits 1 and 2 | Rash, fatigue, headache, retro-orbital pain, neck/ear pain, sore throat, breathing difficulty, cough, expectoration, gastrointestinal signs (nausea/vomiting, diarrhoea, abdominal pain and so on), haemorrhagic signs (nose/gum bleeding, ecchymosis, petechiae and so on), signs of shock (cyanosis, capillary refill), arthralgia, myalgia, loss of appetite, jaundice and so on |
| Medical history | Previous dengue-related or other flavivirus infection as well as vaccination history (self-report) | Previous dengue infection and related hospitalisation<br>Previous infection to other commonly circulating arboviral infection in the area (ie, Yellow fever vaccination history) |
| Laboratory findings | Records from the routine laboratory tests widely used in clinical fever/dengue patient management, as part of the hospital care procedure | Platelet count, haematocrit, haemoglobin, leucocytes, neutrophils, protein level, AST, ALT, urine test results and so on |
| Clinical diagnosis | Clinician's diagnosis with or without referring to the RDT | Diagnosis given by the physician based on clinical presentation after physical examination of the patient |
| Dengue testing results | Results from the dengue tests, mainly RDTs for dengue as well as other commonly circulating arbovirus in the area | Dates of blood draw<br>Test results of the RDT<br>IgM/IgG capture ELISA results<br>PCR results (if available) |
| Treatment | Medicine(s) prescribed and the starting and end dates | Antibiotics, paracetamol, ibuprofen, aspirin and others that may be site-specifically prescribed |
| Outcome | Outcome of this particular visit | Hospitalised, returned home or referral |
| Hospitalisation | Information collected only among hospitalised patients in the surveillance to record other severe signs and progression of illness | Admission and discharge diagnoses<br>Presence of haemorrhagic signs or shock syndrome |
| Hospital charges | Expenses and hospital charges incurred by patient on the visit 1 | Amount of the out of pocket payment by the patient or the family/or guardian<br>Breakdown of the hospital charges (laboratory, medication, admission-related charges) |
| Final outcome | Outcome of the patient's illness at the second visit | Final diagnosis given for the patient, outcome of illness<br>Completion of study participation (early termination and the reason and so on) |

ALT, Alanine AminoTransferase; AST, Aspartate aminotransferase; IPD, Inpatient department; OPD, Outpatient department; RDT, Rapid Diagnostic Test.

patients with laboratory-confirmed dengue by serology and RT-PCR, adjusting for possible confounders, such as age, days since onset of fever, primary versus secondary infection, inpatient versus outpatient and so on. Differences in symptomatic complex of dengue fever (DF) (and DHF, if data allows) by age and serotype will be also determined using multivariable logistic regression.

As outpatient disease accounts for the greater part of dengue disease burden, clinical profile of individuals with DENV infection will be characterised by the type of treatment (hospitalised vs outpatients) as well as by severity of the disease (severe vs non-severe by the 2009 WHO criteria).[46] Classification is determined after the course of illness is completed (typically during the convalescent visit). Symptomatic dengue is classified as outpatient or hospitalised. Progression of dengue is recorded as DF, DHF I, DHF II, DHF III or DHF IV, and clinical patterns will be compared by the severity grade.[46 47] These will be compared with results obtained from other DVI studies in Latin America (Colombia) and Asia (Thailand, Vietnam and Cambodia). Overall, comparisons will be made across Burkina Faso, Gabon and Kenya.

With the age-stratified sera that reflect the age distribution of the general population of the country, the serological survey sampling strategy ensures sufficient subjects to obtain precise age-specific estimates of seropositivity and seroconversion of the catchment area population. The seroconversion rate and change in the immune status will be determined by age group during the study period. The age-stratified serosurvey data will also allow calculation of the force of infection of dengue in the study population. After enrolment, there are subjects who drop out in the follow-up bleeds about 6 months later. Basic demographic information will be compared between those that completed participation and those with incomplete participation to check whether study subjects represent the catchment area population. Comparisons will be made among Burkina Faso, Gabon and Kenya.

### Ethical considerations

To minimise inconvenience of the study to patients, clinicians and nurses were sensitised and trained regarding the study requirements and procedures in order for data collection to be integrated into routine patient care. The clinicians and nurses selected for the study receive coordinated support from study field staff throughout the study process. Written informed consent and assent for participants 7 (13 for Kenya)–17 years of age were obtained from patients by study staff. Study staff go through consent and assent documents for short summary of the disease, detailed description of study procedures and information on reimbursement. Patient data are documented in the study designated office; only the study staff have access to the data that are de-identified. Data are exclusively handled in the study office and stored safely in a protected database in the study office as well as on the DVI main server.

### DISCUSSION

Dengue cases have been detected since the 1960s in Africa, and there has been continued presence of *Aedes* vectors in the continent.[5 7] However, very few dengue studies have been conducted in Africa, and little evidence is based on population-based studies.[6] Compared with the volume of evidence from SE Asia and the Americas, there is critical data scarcity on dengue in Africa. Suspicion of substantial dengue burden in Africa is based on limited reports of outbreaks and a handful of seroprevalence studies testing different viruses among samples that likely do not represent the general population. In the three countries selected for our field studies, somewhat more data are available, but are still very limited. In Burkina Faso, a recent observational study conducted in 2013 reported that 8.7% of the febrile patients showed positive results on dengue RDT.[16] In Gabon, one study suggested minimal DENV circulation in rural areas,[21] while another study reported 12.3% seroprevalence, by IgG antibodies against dengue, among toddlers 30 months of age in semirural parts of Lambaréné.[20] In Kenya, about 13% of

the individuals in Mombasa have been reported to have evidence of past or current DENV infection by RT-PCR and IgM antidengue ELISA after the 2013 outbreak.[26] Despite the limited scope and generalisability of these studies, they suggest that there may be more dengue than previously appreciated due to underestimation and misdiagnosis.[25 26]

These studies suggest the presence of dengue and some level of underlying seroprevalence in the countries of our field studies. However, often these studies are limited by their retrospective design or sample collection (blood donors or sample collected from surveys of other diseases) to demonstrate the true, population-based, burden of dengue. We proposed to address this gap by population-based dengue surveillance and seroprevalence studies in West, (West-) Central and East Africa.

The present studies at three sites in Africa will provide important information on undocumented DENV circulation in Africa. Such data will help to strengthen the evidence base for dengue burden in Africa. Better defined disease burden data based on our studies could be used to assess the relative need for dengue prevention and control measures, such as whether a dengue vaccine would be a cost-effective public health intervention for countries in Africa. Clinical findings from our studies could also be used as a guide for dengue case detection and case management.

The studies have some important limitations. We recognise variability of dengue epidemiology over time and by region. Due to resource constraints, our studies are limited in terms of time frames and geographical extent. These constraints may limit the generalisability of our study results.

One potential source of bias in estimating the incidence of symptomatic dengue is under-ascertainment due to the community residents with relevant symptoms seeking care from other healthcare providers and facilities than the study facilities. As the study design remains passive surveillance, cases are ascertained only at our study facilities. By estimating the proportion of febrile patients seeking care elsewhere as well as refusal rates among the potential enrolees that were screened for eligibility criteria, the degree of febrile patients missed by the study can be determined. Inverse probability weighting will be used to account for these potential subjects missed by the surveillance as adjustments in incidence calculation. Also, depending on the transmission volume of dengue or other cocirculating diseases with onset of fever, there may be patients that are diagnosed with other diseases and ruled out for dengue. Furthermore, with respect to dengue diagnostics for our serological surveys, there are other circulating flaviviruses in Africa leading to challenges in identifying antibodies to past dengue infections. While our testing plan assesses for some flaviviruses, others known to circulate in Africa, such as Banzi and Usutu viruses, are not part of the testing plan.[48–50] Due to resource limitations, serological testing will be limited to yellow fever virus, West Nile virus, Zika virus and Japanese

Encephalitis virus as well as DENV 1–4. Therefore, in some cases, it may be difficult to determine prior exposure to DENV versus other flaviviruses based on serological data. This cross-reactivity may lead to overestimation of dengue force of infection.

In addition, the serosurvey and healthcare utilisation survey are conducted on a randomised subsample of the catchment area population and there may be limited generalisability of the data collected from these surveys. With unknown differences among those that agree to participate and those that do not agree, the data may not be representative of the general population of the study countries.

## CONCLUSION

The data collected from our studies will contribute to the assessment of the unknown dengue disease burden in Burkina Faso, Gabon and Kenya. These data can fill a gap in undocumented burden of dengue in the region and, collectively, may be used to infer dengue burden in other areas of Western, Central and Eastern Africa. Countries in Africa may not consider introduction of a dengue vaccine as a priority in the near future due to many other competing public health problems and limited resources. For cost-effective implementation of public health interventions, accurate data on dengue burden from epidemiological studies would be needed for policy makers to make evidence-based decisions on control and prevention of dengue. Our studies will provide some much needed information based on population-based research to assess dengue burden in Africa.

**Author affiliations**
[1] Global Dengue and Aedes-transmitted Diseases Consortium, International Vaccine Institute, Gwanak-gu, The Republic of Korea
[2] Faculty of Epidemiology and Population Health, London School of Hygiene and Tropical Medicine, London, UK
[3] Epidemiology, Biostatistics and Occupational Health, McGill University, Montreal, QC, Canada
[4] Development and Delivery, International Vaccine Institute, Gwanak-gu, The Republic of Korea
[5] School of Public Health, University of Montreal, Montreal, Quebec, Canada
[6] Centre de Recherches Médicales de Lambaréné, Fondation Internationale de l'Hôpital Albert Schweitzer, Lambaréné, Gabon
[7] Department of Communicable Disease Prevention and Control, Ministry of Health, Nairobi, Kenya
[8] Institute of Tropical Medicine, University of Tübingen, Tübingen, Germany
[9] Eastern and Southern Africa Centre of International Parasite Control (ESACIPAC), Kenya Medical Research Institute, Nairobi, Kenya
[10] Program Equité, Action-Gouvernance-Integration-Reinforcement, Ouagadougou, Burkina Faso
[11] UMI Résiliences, Institut de recherche pour le developpement (IRD), Paris, France
[12] Centre Muraz, Bobo Dioulasso, Hauts Bassins, Burkina Faso

**Acknowledgements** We thank the doctors and laboratory staff of CSPS of Ouagadougou, ASH, Ganjoni dispensary, Tudor subcounty Hospital and Coast Provincial General Hospital as well as at Centre Muraz, CERMEL and KEMRI. We thank collaborators at AGIR, IRD France and University of Montreal, and University of Tubingen. Last, we would like to thank the DVI team as well as statisticians, laboratory, and administrative staff at the International Vaccine Institute for their helpful comments during the preparation of this manuscript and support during the studies.

**Contributors** JKL designed the study, is overseeing data collection and was a major contributor in writing the manuscript. MC codesigned the study, oversaw some parts of data collection and supported in writing of the manuscript. JSL was a contributor in designing of the study and oversight of parts of data collection. KSL was a contributor in oversight of data collection. SN, SKL, EA, NO and AB supported in data collection. VR supported in designing of the study and was a major contributor in finalisation of the manuscript. JF was a contributor in data collection. BL was a contributor in designing of the study and data collection. SHM was a contributor in designing of the study and site establishment. ME was a contributor in designing of the study. EB supported in data generation. SMN was a contributor in designing of the study and site establishment. STA and SY were contributors in designing of the study and site establishment. NA was a major contributor in providing oversight of the data collection and finalisation of the manuscript. IKY was a major contributor in designing of the study and finalisation of the manuscript. All authors read and approved the final manuscript.

**Funding** The current study was supported by funds from the Bill and Melinda Gates Foundation (OPP 1053432). NA receives support from the United Kingdom Medical Research Council (MRC) and Department for International Development (DFID)(MR/K012126/1). VR holds a CIHR-funded Research Chair in Applied Public Health (CPP-137901).

**Disclaimer** The funding body had no role in the design of the study and collection, analysis and interpretation of data and in writing the manuscript.

**Competing interests** None declared.

**Patient consent** Obtained.

**Ethics approval** The protocol for each study obtained ethical approvals from the Institutional Review Boards (IRBs) of the International Vaccine Institute, the London School of Hygiene and Tropical Medicine and the Ethics Committee of host country institutions, including KEMRI Scientific and Ethical Review Unit in Kenya, Gabon National Ethics Committee and Institutional Ethics Committee, Scientific Review Board of CERMEL in Gabon and the IRB of Centre Hospitalier de l'Universitéde Montréal (CRCHUM) at University of Montreal and the National Health Ethical Committee of Burkina Faso.

**Provenance and peer review** Not commissioned; externally peer reviewed.

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
