## [Reviewer comments · BMJ Open]

ARTICLE DETAILS

TITLE (PROVISIONAL)	Evaluating dengue burden in Africa in passive fever surveillance and sero-prevalence studies: protocol of field studies of the Dengue Vaccine Initiative
AUTHORS	Lim, Jacqueline; Carabali, Mabel; Lee, Jung-Seok; Lee, Kang Sung; Namkung, Suk; Lim, Si-Ki; Ridde, Valéry; Fernandes, Jose; Lell, Bertrand; Matendecheo, Sultani; Esen, Meral; Andia, Esther; Oyembo, Noah; Barro, Ahmed; Bonnet, Emmanuel; Njenga, Sammy; Agnandji, Selidji; Yaro, Seydou; Alexander, Neal; Yoon, In Kyu

VERSION 1 – REVIEW

REVIEWER	Scott Halstead Uniformed Services University of the Health Sciences\ No Competing Interest
REVIEW RETURNED	29-May-2017

GENERAL COMMENTS	This is a description of a proposed research project designed to measure the burden of dengue illness in selected areas of three African countries. The authors' review of human dengue experience in Africa is very incomplete. Some of the information cited in references should be summarized in the Introduction supplemented by a series of studies at the University of Ibaden in the 1960-70s, studies on sylvatic dengue 2 in West Africa with ancillary search for infections of humans, dengue outbreaks in West Africa, Cape Verde and Mauritius, and the very substantial and severe and classical dengue hemorrhagic fever that has affected children in Eritrea and Sudan. It is not clear to this observer how the publication of this planned but inadequately referenced research will benefit either the authors or readers. The authors should note that an African dengue resistance gene has been described.¹ The implications of this discovery should be included in the design of surveys for dengue illness in African populations. Specific comments: P 7, line 31 and p 26, line 22. The authors should use the original reference, not one to a review article. The outbreak that occurred in "Africa" in 1823, may actually have been chikungunya. 2,3 Aedes aegypti was described as of African origin by a member of the Linnaeus group in 1762. P 13, line 44. "Expansion factors' for SE Asian children, presumably start with an index case, i.e., a reported hospitalized child and an attempt to estimate how many additional clinically overt illnesses of all degrees of severity have occurred. Are the index cases in Africa hospitalized children? If not, how will
--

	an Asian expansion factor be applied to African data/ P 26, line 35-51. All descriptions of dengue incidence in Africa should be described consistently. What does "incidence" mean? Clinical disease? Hospitalized? Out-patient? If so, what age group, where and when?  1. Sierra B, Triska P, Soares P, et al. OSBPL10, RXRA and lipid metabolism confer African-ancestry protection against dengue haemorrhagic fever in admixed Cubans. PLoS Pathog 2017; 13(2): e1006220. 2. Christie J. Remarks on "kidinga Pepo" a peculiar form of exantematous disease. Epidemic in Zanzibar, East Coast of Africa, from July 1870 till January 1871. BMJ 1872; 1: 577-79. 3. Halstead SB. Reappearance of chikungunya, formerly called dengue, in the Americas. Emerg Infect Dis 2015; 21(4): 557-61.
--	---

REVIEWER	Natalie Dean Department of Biostatistics, University of Florida, Gainesville, FL, USA Former WHO consultant in charge of a technical working group on the design and conduct of dengue serosurveys to inform vaccination.
REVIEW RETURNED	06-Jul-2017

GENERAL COMMENTS	Overall the study will provide valuable information about dengue burden and health-care seeking behavior in a region with sparse data. The individual components of the study each have a unique and useful role. I would recommend that this protocol be published pending revisions. I have answered no on three of the items above (sufficient description of method, statistics, and English) to flag your attention to my comments below. I felt that some of the descriptions of the sampling procedures, sample size calculations, and statistical analysis were vague and would benefit from clarification. Major comments:  - ABSTRACT. Fix the English in "Strengths of this study" and "Limitations of this study" sections. - Page 8, Line 47: Temper the statement that you are measuring the true burden of dengue in West, Central, and East Africa, as you have clearly stated that sites were selected by convenience. Could rephrase as measuring the true burden of dengue *in selected sites* from each of these three regions. - Page 14, Line 35. It is strange that you have different inputs for each site but end up with the same sample size. Please explain the process. - Page 24, Line 11. Here it is stated that the numerator will be dengue cases in eligible individuals. As consent for participation is an eligibility criteria, how will refusal rates for study participation be handled? If refusal rates were high but the denominator is still the total population size, this would induce a downward bias in estimated incidence. This may be addressed in the following paragraph when discussing
---

adjustment for cases missed by passive surveillance, though the text describes only a “comparison” between consenters and non-consenters rather than an “adjustment”. The language about the proposed comparison is unclear.

- Page 24, Line 18. How will the variance be inflated to account for the proposed under-reporting adjustments?

- Page 25, Line 11. There is insufficient discussion of how the serosurvey results will be estimated. These are household based surveys with multi-stage sampling, involving clustering and stratification. For example, on Page 18, you describe density-dependent sampling of pre-selected households until a cap is reached, with random sampling of eligible individuals within the households to achieve a given age distribution. Non-response weights may be necessary to account for participation refusal. Attention must be paid to how the point estimates and variance are estimated to properly account for these design features.

- Page 25, Line 15. What will the age-specific estimates look like? 5 year bands? To claim you have sufficient precision for age-specific estimates requires more justification. Looking back at the sample size calculations on Page 14, it appears to calculate a single sample size for the entire population, so you would have significantly lower precision when subdividing the ages. Furthermore, clustering of individuals by household or neighborhood may reduce the efficiency of your estimation. It is not clear to me that this is considered in the sample size calculations either.

Minor comments:

- Why don't the line numbers line up with the lines?

- Page 9, line 22. “A recently published study results demonstrated” -> “Results of a recently published study demonstrated”

- Page 11, line 15. “Positioned at the fourth country” -> “Positioned as the country with the fourth fastest”

- Page 25, line 11. “reflect [the] age distribution”

REVIEWER	Thomas Jaenisch Section Clinical Tropical Medicine, Department for Infectious Diseases Heidelberg University Hospital 69120 Heidelberg, Germany
REVIEW RETURNED	11-Jul-2017

GENERAL COMMENTS	The authors describe the study protocol for evaluating the dengue burden in Africa through passive fever surveillance and sero-prevalence studies. This is a well-written protocol manuscript and it deserves to be published. This reviewer only has minor comments – with the exception of one methodological comment regarding site selection: - Why did the authors not make the presence of an HDSS a mandatory site selection criterion – OR: which information is available in Lambarene and Mombasa that satisfied the authors with regard to the denominator data of the populations involved? Minor comments: Background: - The Nature paper by Bhatt et al., is mentioned (ref#1) with regard to the estimated incidence of 50-100 Mill. symptomatic infections – but not with regard to the fact that the estimated number of infections in Africa equals that of Latin America. This would enrich the manuscript, providing the argument that “modelling suggests a high burden, but we have to validate it with empirical data”. Methods: - The citation of Brady et al. (ref#9 in the methods section) is not well-placed as this is not strictly a modelling paper, but rather on consensus evidence. It shows the uncertainty of the consensus estimates in Africa about the presence of dengue. - RE Fever surveillance – laboratory testing (PCR): Was PCR carried out in all three sites locally? If yes, how was the quality of PCR testing standardized? - RE Serological survey – laboratory testing: How big is the subset that will undergo PRNT / Luminex testing?
---

VERSION 1 – AUTHOR RESPONSE

-Reviewer 1, comment #1:

Changes are reflected in the body of the manuscript.

-Reviewer 1, comment #2:

We rephrased our statement and clarification has been made in the body of the manuscript.

-Reviewer 1, comment #3:

The description of the methods was presented in the body of the manuscript. The description of 'incidence' was presented in the "Planned statistical analysis" sub-section, now highlighted in yellow for your reference.

-Reviewer 1, comment #4:

The list of reference has been updated accordingly.

- Reviewer 2, comment #1

Changes have been made in the abstract section

- Reviewer 2, comment #2

Changes are reflected in the body of the manuscript

- Reviewer 2, comment #3, about the sample size

It is correct that we used different inputs for each site. As we indicated on Page 14, Line 35, we used different non-response rates by site. So, for example, for the Gabon site, we used 10% non-response rate to reach the final sample size 3,000. For Kenya and Burkina Faso where we had higher seroprevalence estimates available, we had smaller sample sizes calculated. We used higher non-response rates and rounded-up to reach 3,000. We had conducted the same set of studies in Asia and L. America and experienced losing enrolled subjects in the follow-up bleeds occurring about 6 months later. So, after consultation with the local collaborators, we applied different non-response rates and agreed to enroll 3,000 subjects for the serosurveys.

- Reviewer 2, comment #4

The statement was rephrased for clarity. Information about methodological tools to overcome the issue of censoring (refusal, loss to follow-up etc.) has been added.

- Reviewer 2, comment #5, about the variance to be inflated

This was addressed in the 'planned statistical analysis' section of the methods section. However, additional description on how we will handle this variance is shown as changes in the discussion section.

- Reviewer 2, comment #6, about sampling of the serosurvey and how the results will be estimated

In the serosurvey, one of the inclusion criteria was the subject's ability to provide paired sample. Paired samples are necessary to perform the necessary dengue diagnostics and for lab confirmation. Thus, the analysis sample will only include those subjects who had complete participation over multiple bleeds.

For sample size calculation, we used one seroprevalence estimate found in the literature for one particular age group (usually children) for each of the countries to calculate sample size across all ages. There were no other prevalence estimates reported or estimates from multiple age bands. Prevalence will only increase with age and, if we had used a higher prevalence estimate (e.g., what

we may find for older adults, if data were available), then the sample size would have been smaller than what we used. Therefore, for all three sites, we had conservative estimates (i.e. bigger sample size than what may be needed). In addition, we applied 10-30% of non-response rates. With our conservative estimates for the sample size and non-response rates applied, we have a large enough sample size in each site. Given the limited evidence on dengue prevalence in the study areas, we allowed acceptable level of difference (marginal error) between the true population and our sample population, to have the resulting estimate to fall within 25% of the proportion (from the available seroprevalence data). Therefore, when we back-calculated using preliminary results, we found that the difference between a true population proportion and our sample population proportion is less than the marginal error, which satisfies the assumption used for our sample estimation.

- Reviewer 2, comment #7, about precision for age-specific estimates

For these household surveys, while it was true that we referred to the previously collected GIS information of the households in Ouagadougou for the site in Burkina Faso, such information was not available for the sites in Kenya and Gabon. Sampling involved stratification where 45% of the samples were collected from children 1 - 14 years-of-age, and 55% were targeted to be collected from adults between 15 and 55 years of age. This was to reflect, very roughly, the age distribution of the general population. If it was narrowly divided (i.e. 5 year age bands), it may not be too feasible for the study team as they go for enrollment.

It is correct that we had a single sample size. As explained above for comment #6, we used seroprevalence of one particular age group (from the published data, mostly from children) to calculate sample size for all age groups. Given that sero-prevalence increases with age, we made conservative estimates (using a sero prevalence estimate from younger age group for all age groups, resulting in higher sample size than what we would have reached if we used a higher sero-prevalence) and, in addition, applied non-response rates (10-30%). We have large enough sample size in all age groups, even when we divide the single sample size with 5-year bands. We wish to generate age-specific estimates with 5 year age bands.

Regarding the age groups, as the analysis progresses, if we find that sample sizes are smaller in particular age groups (fewer enrollees than the sample size after subdividing the ages), we will consider weighting, as needed.

Also, we agree with the reviewer that clustering of individuals by household or neighborhood would reduce the efficiency of the estimation. However, we would have to review the data to see how many of the individuals were from same households (as it was voluntary to participate when we made household visits). Also, we would need to check the distribution of residents by neighborhood to see if our sample is representative after we decide on the final analysis sample (with complete participation).

- Reviewer 3, comment #1, about use of the HDSS availability in site selection

It would have been ideal to use availability of HDSS as one of the priority criteria for site selection. However, we were to choose one site in the western, central, and eastern parts of the continent. In addition to Ouagadougou, Burkina Faso which already had HDSS, we chose Lambaréné, Gabon, for the central part, based on some published information on dengue and their well-established research infrastructure. For the eastern part, we chose Kenya, also based on some of published data and existing research infrastructure. Kenya also has HDSS in five locations in the country. However, when we consulted our local collaborators, considering the sample size calculated, local interests for dengue, etc., it was decided for the studies to take place in Mombasa.

- Reviewer 3, comment #2

Background: Changes have been made in the body of the manuscript.

- Reviewer 3, comment #3

The list of references has been updated accordingly.

- Reviewer 3, comment #4

Samples were shipped to the Clinical Immunology Laboratory of IVI. Detailed clarifications have been made in the body of the manuscript.

-Reviewer 3, comment #5

Flow cytometry-based neutralization assays will be performed against selected flaviviruses in about 50 samples per bleed for 4 bleeds in Burkina Faso and 2 bleeds in Gabon and Kenya.

A Luminex-based multiplex immunoassay will be performed on a randomly selected sub-sample to assess for IgG to different flaviviruses in about 200 samples per bleed for 4 bleeds in Burkina Faso and 2 bleeds in Gabon. Details are presented in the body of the manuscript.

I hope this letter provides further details needed to clarify those issues raised by the reviewers. If you have any questions, please do not hesitate to contact me. Thank you.

VERSION 2 – REVIEW

REVIEWER	Scott B. Halstead Uniformed Services University of the Health Sciences, USA No Competing Interest
REVIEW RETURNED	14-Aug-2017

GENERAL COMMENTS	This manuscript describes an on-going prospective study designed to identify the etiology of febrile cases seen in clinical facilities in three African countries. The investigators will test random subsets of febrile patients from five clinical facilities for 2 years in Ougadougou, Burkino Faso, in one facility in Lambarene, Gabon and three facilities in Mombasa, Kenya, each for a year and one-half. In addition, dengue antibody prevalence and infection rates will be measured in randomly sampled open populations over an interval of 6 months each in Lambarene and Mombasa and at three different intervals over 3 years in Ougadougou. Dengue antibody population prevalence will be measured by indirect dengue IgG ELISA with limited sub-studies on neutralizing antibodies. The investigators should be more circumspect in their claims. Dengue epidemiology, carefully studied in multiple sites over a half century, is characterized by periods of annual high, low and no transmission. The very limited time frames for measuring acute illnesses or dengue virus infections will not permit generalizations of any kind. Further, the investigators badly underestimate the complexity in Africa of identifying human antibodies that can be attributed to past dengue infections. Seroprevalence studies on populations in sub-Saharan Africa have been geographically focal and directed at a limited number of viruses. Here, they will test sera for neutralizing antibodies to yellow fever, West Nile, Zika and Japanese encephalitis (JEV). Despite a recent report of isolation of JEV, there is no evidence that JEV is enzootic
--

	in Africa. There are other flaviviruses that circulate in Africa. These should be identified and included in analyzing serosurvey data. Specific comments: Background, 2nd para. The 1779-80 outbreak of “dengue” in Africa has been identified by experts as an outbreak of chikungunya, identified historically using the name “dengue.” Chikungunya arrived in the American hemisphere in 1827. [1, 2] Selection site. Measuring the incidence of dengue infections in selected facilities in three cities in Africa will not provide a “measure [of] the true burden of dengue.” It is not clear that it is possible to determine the “true” burden of dengue anywhere at any time. Better say “estimate the burden of dengue” and leave it at that. More universal data require samples from a wide spectrum of the total population representing many geographically and ecologically different observation sites. P 9. Under-reporting. The issue of “under-reporting” is frequently mis-characterized. Some individuals estimate the failure to report mild disease as “under-reporting.” Others, estimate that hospitalized cases are under-reported. Which of these different kinds of “under-reporting” are being discussed here? [1] Carey DE. Chikungunya and dengue: A case of mistaken identity? J Hist Med Allied Sci. 1971;26:243-62. [2] Halstead SB. Reappearance of chikungunya, formerly called dengue, in the Americas. Emerg Infect Dis. 2015;21:557-61.
--	--

REVIEWER	Natalie Dean, Assistant Professor of Biostatistics Department of Biostatistics, University of Florida, Gainesville, FL, USA
REVIEW RETURNED	28-Aug-2017
GENERAL COMMENTS	The authors have adequately addressed my earlier comments. I believe the study results will be useful and informative for better understanding the epidemiology of dengue in Africa.

VERSION 2 – AUTHOR RESPONSE

-Reviewer 1, general comment:

We understand that dengue epidemiology is characterized by periods of annual high, low and no transmission and recognize that our projects are limited by study periods which only cover certain time frames. We understand that this may limit the extent of generalizability of our data. However, there were resource constraints and limiting factors when determining study period, study areas, as well as lab tests as to which viruses would be included in our testing plans. More on this is included in the discussion section of the manuscript (highlighted in yellow).

-Reviewer 1, Specific comments:

We appreciate comments from the reviewer and changes are reflected in background, methods, and the body of the manuscript (see highlights in yellow).

About under-reporting, we clarified under-ascertainment (missing those subjects that do not seek care at our facilities) and under-reporting (failure to correctly diagnose the case and to report the case) in the body of the manuscript.

Thank you very much for your consideration. Please do not hesitate to contact if there are any other questions.

VERSION 3 – REVIEW

REVIEWER	Scott Halstead Uniformed Services University of the Health Sciences, USA No Competing Interest
REVIEW RETURNED	12-Sep-2017
GENERAL COMMENTS	This manuscript is a revised manuscript describing prospective studies on dengue virus transmission and disease in human populations living in three sites in Africa. Specific comments: P 3, line 28. Serological prevalence of dengue – insert, “antibodies” Was this a survey of adults and children proportional to their distribution in the population?

VERSION 3 – AUTHOR RESPONSE

Reviewer: 1

Reviewer Name: Scott Halstead

Specific comments:

P 3, line 28. Serological prevalence of dengue – insert, “antibodies”

Was this a survey of adults and children proportional to their distribution in the population?

Response: What has been changed in the manuscript in response to the above comment is reflected in the body of the manuscript. This change is shown on page 6 of the marked copy in yellow highlights.

In response to the question raised by the reviewer asking whether a reference we used (on page 6) was from a survey of adults and children proportional to their distribution in the population, we would like to clarify that the article we referenced was not based on a survey of adults and children proportional to their distribution in the population, but was based on a survey among pregnant women and blood donors from rural (Nouna) and urban (Ouagadougou) Burkina Faso. There were 683 samples tested in total. While it was not based on age-stratified sera, this article provides limited, but locally-acquired, data on dengue seroprevalence from Ouagadougou, Burkina Faso.

For further clarity and to avoid confusion, we have made minor edits throughout the manuscript (see track-change). Thank you very much for your consideration.

VERSION 4 – REVIEW

REVIEWER	Scott Halstead Uniformed Services University of the Health Sciences No Competing Interest
REVIEW RETURNED	16-Oct-2017
GENERAL COMMENTS	Please proceed.